# Cautionary Observations Concerning the Introduction of Psychophysiological Biomarkers into Neuropsychiatric Practice

Paul E. Rapp [1,*], Christopher Cellucci [2], David Darmon [2] and David Keyser [1]

1 Department of Military and Emergency Medicine, Uniformed Services University, Bethesda, MD 20814, USA; david.keyser@usuhs.edu
2 Aquinas LLC, Berwyn, PA 19312, USA; cellucci@gmail.com (C.C.); david.m.darmon@gmail.com (D.D.)
* Correspondence: paul.rapp@usuhs.edu

**Abstract:** The combination of statistical learning technologies with large databases of psychophysiological data has appropriately generated enthusiastic interest in future clinical applicability. It is argued here that this enthusiasm should be tempered with the understanding that significant obstacles must be overcome before the systematic introduction of psychophysiological measures into neuropsychiatric practice becomes possible. The objective of this study is to identify challenges to this effort. The nonspecificity of psychophysiological measures complicates their use in diagnosis. Low test–retest reliability complicates use in longitudinal assessment, and quantitative psychophysiological measures can normalize in response to placebo intervention. Ten cautionary observations are introduced and, in some instances, possible directions for remediation are suggested.

**Keywords:** biomarkers; electroencephalography; event-related potentials; heart rate variability; diagnosis; sensitivity; specificity

## 1. Introduction

Psychophysiology is the branch of physiology dealing with the relationships between physiological processes and psychological phenomena (thoughts, emotions, and behaviors). Clinical psychophysiology is more narrowly defined here as the use of psychophysiological measures to inform assessment and treatment of neuropsychiatric disease. At present, a substantial body of literature has identified psychophysiological measures that are different in clinical versus control populations. Examples will be described. Three applications are critical to the introduction of these results into clinical practice: diagnosis, longitudinal assessment of treatment response or disease progression, and identification of individuals in the subsyndromal state who are at risk of neuropsychiatric disorders. The possibility of combining statistical machine learning technologies with psychophysiological measures to address these objectives has appropriately generated a great deal of enthusiasm. It is argued here, however, that this enthusiasm should be tempered with an appreciation of the challenges that are still ahead. The objective of this study is the presentation of ten cautionary observations. Indications of how some of these challenges might in part be addressed are also presented. The following observations are addressed:

1. Though frequently unreliable, patient report is, and will remain, central to clinical practice;
2. Distortions of cognitive processes can be an element in some neuropsychiatric presentations, and the physiological implementation of these processes is not understood;
3. The interaction of conscious and unconscious processes is not understood, but is clinically important;
4. Psychophysiological measures are characterized by broad distributions;
5. Psychophysiological measures have low diagnostic specificity;

6. The test–retest reliability of psychophysiological measures is frequently untested and can be unacceptably low;
7. Psychophysiological measures vary with age, sex, and ethnicity, thus complicating determination of normative values and reliability;
8. As the result of central nervous system adaptation rather than repair, psychophysiological measures do not invariably normalize during recovery;
9. Psychophysiological measures can change and, in some cases, normalize in response to placebo interventions;
10. The mathematical procedures of statistical learning are not robust to misapplication and to data artifacts.

## 2. The Central Role of Patient Report

*2.1. Observation*

The limitations of patient self-report are commonly recognized [1,2]. Patients can give inadequate report due to denial, lack of insight, or willful intention to mislead. This has motivated the introduction of psychophysiological measures into clinical practice. Additionally, psychophysiological measures are by definition related to physiological processes and this physiological information may inform treatment. Nonetheless in neuropsychiatric practice, patient report remains critical to all that follows. This is arguably true in all areas of practice, but it is particularly true in neuropsychiatry, where there are no dispositive clinical measures analogous to blood pressure, plasma glucose concentration, or tumor volume. While physiology may be part of the story, it is not all of the story. Patient report remains the final arbiter.

*2.2. Response to Observation*

Recognizing that patient reports are integral to practice, an effort can be made to obtain these reports systematically with standardized questionnaires [3]. Constructing and establishing the reliability and validity of a health scale is a significant undertaking [4]. This argues strongly for the use of previously developed and validated questionnaires. In the case of clinical trials, the US Food and Drug Administration has provided guidance [5]. More generally, the COSMIN study produced an international consensus on measurement properties for health related, patient-reported outcomes (COSMIN, Consensus-based Standards for the Selection of a Health Measurement Instrument, [6–8]). This panel generated a checklist for assessing measurement instruments [9,10]. The COSMIN criteria assess three elements of reliability, seven elements of validity, responsiveness, and interpretability. It is suggested that investigations use questionnaires that satisfy COSMIN criteria.

## 3. Distortions of Cognitive Processes Can Be an Element in Some Neuropsychiatric Presentations, and the Physiological Implementation of These Processes Is Not Understood

*3.1. Observation*

We do not understand the physiological basis of cognitive processes such as attention, perception, memory, language, communication, decision-making, and the implementation behaviors. Indeed, some have argued that this understanding is not possible ([11]; for an alternative view, however, see [12]). Neuropsychiatric presentations can include deformations of cognition ranging from minimal, in cases of mild cognitive impairment, to profound, in some presentations of psychotic disorders. This has clinical implications. As summarized by Harrington [13]: "After all, current brain science has little understanding of the biological foundations of many—indeed most—everyday mental activities. This being the case, how could current psychiatry possibly expect to have a mature understanding of how such activities become disordered—and may possibly be reordered". We have previously observed that twenty-first century clinicians are charged with constructing a physiologically-informed response to these presentations. An understanding of the physiological basis of higher cognitive processes is therefore not simply a matter of deep scientific

and philosophical importance. Rather, it is of immediate clinical significance [14]. While the ultimate accessibility of an understanding of the basis of cognition is a matter of debate, it is certainly true that a solution is not available at present.

### *3.2. Response to Observation*

While the physiological implementation of cognition is not understood, a great deal is now known about its neural correlates [15–17]. Examples of neural correlates of cognitive processes can be introduced best by considering specific processes. Event-related potential (ERP) experiments have been constructed to identify electrophysiological correlates of specific cognitive processes including selective attention [18], working memory [19], episodic memory [20], language [21], and emotion [22]. The search for ERP-based clinical biomarkers is encouraged by the large body of literature showing alteration of ERPs in clinical populations including depression [23], schizophrenia [24], neurodegenerative diseases [25], dementia [26], post-traumatic stress disorder, PTSD [27], autism [28], borderline personality disorder [29], and generalized anxiety disorder [30].

### 4. The Interaction between Conscious and Unconscious Processes Is Not Understood but Is Clinically Important

### *4.1. Observation*

The mystery deepens when we recognize that unconscious processes are a significant component of psychological functioning [31] and that the physiological basis of unconscious cognitive processes is largely unknown. It should be noted that the current conceptualization of the unconscious does not rely exclusively on the psychoanalytic foundations of dynamic psychology. The present conceptualization has been variously described as the cognitive unconscious [32], the psychological unconscious [33], the adaptive unconscious [34], and the modern unconscious [35]. Bargh has provided a valuable summary statement of current thought: "The elegance of the modern research on unconscious processes is that it combines the best of these three major psychological theories (psychoanalysis, cognitive psychology, behaviorism). What this research reveals is that many important affective, motivational, and behavioral phenomena operate without the person's awareness or conscious intention (Freud) and that they are often triggered by events, people, situational settings, and other external stimuli (behaviorism), but that these external stimuli exert their effect through their automatic activation of internal mental representations and processes (cognitive psychology)" [35].

As in the case of conscious cognitive processes, distortions of unconscious processes can have clinical implications. Emotional processes can be unconscious [36] and unconscious processes can have a significant impact on health [37]. Wiers et al. [38] concluded that "implicit processes might be particularly important in psychopathology". An understanding of the physiological basis of the unconscious is therefore also an unmet clinical requirement.

### *4.2. Response to Observation*

The psychophysiology of the unconscious can also be investigated empirically ([39–43], these are representative examples drawn from a larger literature). As in the case of conscious cognitive processes, quantitative measures developed by these investigations may prove to be of clinical value, but it must be recognized that the methodological challenges of quantifying unconscious cognitive processes are far greater than those encountered in the investigation of conscious cognition.

An early examination of putatively unconscious perception (perception in the absence of awareness) was the investigation of blindsight [44]. Patients with focal damage to the striate cortex can lose conscious perception in a restricted visual field. They can nonetheless report properties of visual stimuli projected to the damaged visual field at above chance levels. In some cases, response accuracy " reaches 90% to 100%" [45]. It was suggested that partial sparing following surgery/stroke/accident could leave localized areas (islands) of

intact function. The ability to track objects moving in the damaged hemifield argues against this possibility. A suggestion of clinical significance is obtained from blindsight results indicating that emotional processing can occur in the absence of conscious perception. De Gelder et al. [46] reported that their participant could correctly distinguish happy versus fearful faces presented to the blind field with 66% accuracy. This study included recordings of event-related potentials. The difference in ERPs elicited by happy versus fearful faces in the intact visual field were similarly observed in ERPs elicited by stimulation to the blind hemifield. Tamietto et al. [47] found that emotionally-valenced stimuli projected to a blind field can elicit a physiological response (pupillary responses and electromyogram responses in the corrugator supercilia (frown muscle) and the zygomaticus major (smile muscle)).

The results obtained in blindsight studies are of interest to the study of the physiological basis of psychopathology because they indicate that emotional processing can occur in the absence of conscious perception. These results are, however, essentially theoretical (proof of concept) and have limited potential for clinical application because they are obtained in a very special population. Event-related potentials obtained with subliminal visual stimuli are potentially of greater utility because they can be readily obtained in control and clinical populations. Song and Yao [48] provide the following summary: "Moreover, the influence of subliminal visual stimulus is not limited to low-level sensory domains but also evident in high-level cognitive domains, where subliminal stimulation of achievement-related words was found to influence goal pursuits and improve task performance". For example, electrophysiological studies of unconscious visual stimulation with emotionally-valenced stimuli conducted with a control population found different event-related potential responses to supraliminal and subliminal stimulation. Liddell, et al. [49] and Kiss and Eimer [50] observed an enhanced N2 ERP component (a negative-going ERP waveform with a maximum amplitude between 200–350 ms after stimulus presentation in response to subliminally-presented fearful faces but in not in response to supraliminal presentation. Kiss and Eimer [50] wrote that the results, "strongly suggest that it (the difference in supraliminal and subliminal responses) reflects a genuine and distinct electrophysiological correlate of subliminal emotional processing".

Of greater immediate interest are those studies that show differences in subliminal responses in control and clinical populations. Del Cul, et al. [51] determined perceptual thresholds in a backward masking experiment with schizophrenic patients and compared the results to those obtained from a control population. They found preserved subliminal processing and impaired conscious processing in the patient population. The thresholds to conscious perception were measured and individual values were correlated with symptom severity. They argue that the control versus patient results are consistent with deficits in late-stage conscious perception.

A comparison of backward masking in control and schizophrenia populations were also reported in Green, et al. [52]. Eleven patients with recent-onset schizophrenia in a period of unmedicated remission were compared against a matched control group in a visual masking experiment. Patients in psychotic remission showed significant deficits. It had been hypothesized that performance deficits in the backward masking task may indicate an underlying predisposition to schizophrenia. Because the patients in this study were in remission, Green et al. argued that "these data from patients in well-documented psychotic remission add converging support for the hypothesis that deficits in backward masking procedures are indications of vulnerability to schizophrenia" [52]. Results of this type may be contributory to the third of our stated objectives of clinical psychophysiology: identification of individuals at risk of disease onset. In a subsequent backward masking investigation, Green et al. [53] obtained electrophysiological recordings of event-related gamma activity (30–35 Hz) from controls and from patients with schizophrenia. The control participants, but not the schizophrenic participants, showed a burst of gamma activity 200–400 ms following stimulus presentation. The authors suggest that this failure of gamma activity may be causatively related to perceptual deficits seen in some schizophrenic patients.

While the examples previously cited in response to Observation 2 and here in response to Observation 3 are encouraging, these results provide a very incomplete understanding of the physiological implementation of conscious and unconscious cognitive processes. Additionally, it should be recognized that the challenges outlined in subsequent sections of this contribution are applicable to these results.

## 5. High Inter-Individual Variation

### 5.1. Observation

The next cautionary observation is admirably summarized by Shackman and Fox in the title of their 2018 contribution [54] "Getting serious about variation: lessons for clinical neuroscience", in which they cite the Holmes and Patrick paper "The myth of optimality in clinical neuroscience" [55]. Psychophysiological measures are characterized by broad distributions. In this context, it is appropriate to emphasize that statistically significant between-group differences do not ensure their usefulness in a classification. The commonly-used independent samples *t*-test assesses the difference in the means of two distributions. The distributions can, however, overlap. A particularly instructive example is given in Holmes and Patrick (Figure 1 of [55]), which presents a measure of frontotemporal connectivity obtained in patients with psychosis and healthy comparison participants. The group means are different, but the distributions show substantial overlap. They conclude: "Analytic approaches that focus on group differences may mask the presence of substantial overlap in phenotypic distributions providing an illusion of diagnostic specificity."

### 5.2. Response to Observation

When considering the potential utility of a measure in a classifier, as noted by Holmes and Patrick, the independent samples t-test is not a completely satisfactory indicator. In the case of a dichotomous classification, rejection of the null hypothesis is not an adequate indication of usefulness in classification. An appropriate indicator is the probability of error in a classification. For example, Rapp et al. [56] have published a computational example in which the *p* value is $p = 2.1 \times 10^{-12}$ but the classification error is 0.408, where the error rate for classifying with equal probability to the two classes is 0.5.

In the case of a dichotomous classification where it can be assumed that the distribution of the adjudicating measure in both groups is normal, or normal-enough, an analytic estimate of the probability of classification error is available [57] (see also [58]). Alternatively, an empirical determination of error can be obtained with a leave-one-out cross validation. The LOOCV has the advantage of generalizing to classification problems that include more than two groups. Determination of classification error rates should be included with the results of a *t*-test when reporting between-group differences observed with a psychophysiological measure.

## 6. Psychophysiological Measures Have Low Diagnostic Specificity

### 6.1. Observation

The low specificities of psychophysiological measures complicate their use in diagnosis and as prodromes used in the identification of at-risk individuals. For example, Rapp et al. ([59] Table 8) have identified results in the literature showing altered EEG synchronization patterns in AD/HD, alcohol abuse, alexithymia, autism, bipolar disorders, dementia, depression, hallucinations, HIV dementia, migraine, multiple sclerosis, Parkinson's disease, post-traumatic stress disorder, schizophrenia, and other psychotic disorders. A similar literature study identified non-specific changes in functional connectivity in eleven clinical conditions ([59] Table 3).

A further complication must be recognized. Sensitivity and specificity results obtained in clinical studies with well-defined participant groups are misleading if not interpreted with care. Results obtained in a study where the comparison is between healthy persons and a "pure" clinical population that has satisfied rigorous inclusion/exclusion criteria

can be unwarrantably optimistic. Specificity is expected to be far lower when a general psychiatric intake population is considered.

An additional consideration merits attention. It could be argued that the discouraging results obtained when psychophysiological measures are used in diagnosis could reflect deficiencies in diagnostic systems, notably the DSM-5. Kapur, Phillips, and Insel [60] have suggested that, in part, a commitment to support DSM-5 diagnostic structures has been an impediment to the development of clinically-useful biomarkers. Newson et al. [61] conducted a quantitative analysis of 107,349 adults using the Mental Health Quotient. Of those participants whose symptoms mapped to at least one of the ten DSM-5 diagnostic disorders considered in the study, the heterogeneity of symptom profiles was almost as high within a disorder as between two disorders, and "not separable from randomly selected groups of individuals with at least one of any of the 10 disorders." In summary they concluded, "Overall, these results quantify the scale of misalignment between clinical symptom profiles and DSM-5 disorder labels and demonstrate that DSM-5 disorder criteria do not separate individuals from random when the complete mental health symptom profile of an individual is considered." This conclusion could suggest that psychophysiologically-based diagnosis in support of the DSM-5 classifications is not possible in principle. The Newson et al. results should, however, be compared with the results of DSM-5 interrater reliability field trials that report either "very good agreement" or "good agreement" in fourteen of twenty diagnostic categories ([62] Figure 1). Details of the field trials are given in [63–65].

The identification of individuals at risk of neuropsychiatric disease is a special case of a diagnostic process. Some positive results have been reported; for example, see Byeon's review of studies predicting high dementia risk [66]. A valuable alternative direction for the identification of individuals at risk of disease onset was presented by Beaudoin and colleagues [67]. Rather than predict disease specific symptoms, they used machine-learning technologies and symptom profiles obtained from schizophrenic patients to predict quality of life as assessed with the Heinrichs–Carpenter Quality of Life Scale. As they report "three models were constructed (1) baseline prediction of 12-month QoL, (2) 6-month prediction of 12-month QoL and (3) baseline prediction of 6-month QoL". They found that the best predictors included social and emotion-related symptoms, processing speed, gender, treatment attitudes, and mental, emotional, and physical health. While encouraging, challenges to identifying individuals at risk should be recognized. To all of the previously described difficulties of psychophysiologically assisted diagnosis we must introduce additional problems, specifically the challenges of a long-term longitudinal study and the very large study populations required. In contrast with other areas of medical practice, reliable predictors of neuropsychiatric disease onset are often unavailable. Obstacles in the implementation of the-much hoped-for arrival of preemptive psychiatry [68] should be recognized. Consider the structure of the study. An asymptomatic or subsyndromal population is identified and psychophysiological measures are acquired on intake. The population is followed for a specified duration, possibly months or years. Participants who remain stable (Stables) and those who present the disorder (Converters) are identified. The Stable versus Converter discrimination can itself be difficult. An attempt is then made to construct a classifier that discriminates between Stables and Converters using baseline intake data. By definition, this is a longitudinal study. The challenges including expense and loss of participants to follow up are well known to investigators with experience in long-term clinical studies.

Additionally, the number of individuals required in a prodrome search is much larger than might be expected. A critical determinant of the required sample size is the conversion rate in the study population. Conversion rates to psychiatric disorders in the general population are typically low. For this reason, investigators will attempt to identify an enriched population with a higher conversion rate. For example, in a study of electrophysiological prodromes of delayed-onset PTSD, Wang et al. [69] followed military personnel who had recently returned from combat. For this population the conversion rate is high, about 10%. Even in an enriched population, the number of required participants can be high. There is more than one procedure for estimating the required sample size to

obtain a confidence interval with a prespecified precision. Wang et al. used Hoeffding's inequality [70]. Using this criterion, the sample size required for $\pm 0.1$ sensitivity estimate with a 95% confidence interval requires 185 Converters. If the conversion rate is 10% a total sample size of 1850 participants may be required. By fixing the precision of the interval estimate independently of the underlying sensitivity, a sample size determination based on Hoeffding's inequality is conservative in the sense of requiring a large sample size. Alternatively, one could determine the sample size required to give an expected width using the Clopper–Pearson interval [71,72]. With this criterion, the required number of Converters for $\pm 0.1$ is 104, giving a total sample size of 1040. Even with this more encouraging sample size, the study sample is significantly larger than that usually observed in the prodrome literature. For small sample size studies, including the report of the confidence intervals of sensitivity and specificity is therefore especially important.

Problems associated with underpowered studies are not limited to identification of prodromes of neuropsychiatric disorders. Button et al. [73] argue that the problem is ubiquitous in neuroscience. They stress that "...it is less well appreciated that lower power also reduces the likelihood that a statistically significant result reflects a true effect. ... The consequences of this include overestimation of effect size and low reproducibility of results."

### 6.2. Response to Observation

The low diagnostic specificity of psychophysiological measures can to a degree be addressed by expanding the analysis to multivariate classification. A well-known example from physiology provides an example. acid-based disorders are broadly classified as alkalotic versus acidotic with respiratory or metabolic etiologies. A discrimination cannot be made by measuring pH alone or bicarbonate alone. Both must be measured simultaneously. Similarly, it is suggested that the utility of computationally-informed neuropsychiatric assessment might be advanced by incorporating multiple psychophysiological measures and, importantly, other classes of measures (patient history, family history, imaging, genetics, epigenetics, etc.). Walk tests, which have been utilized in the assessment of depression [74–76] and schizophrenia [77–79], provide an example of additional measures that when combined with psychophysiological measures might improve diagnostic specificity.

The incorporation of multiple measures introduces an additional challenge: model selection. Predictors used in a classifier must be selected from the set of candidate predictors in a statistically-responsible fashion. More is not necessarily better. Watanabe et al. [80] give a classification example (eyes open versus eyes closed; no task EEGs) where the classification error rate decreases as measures are eliminated from the classifier ([80], Figure 6). Fortunately, an extensive literature exists to guide model selection [81]. The validation of multivariate classifiers is an essential activity. Unfortunately, this is not always done correctly. Attention is directed to Section 7.10.2, "The Wrong and Right Way to Do Cross-Validation" in Hastie, et al. [81] (see also [82]).

It is impossible to measure everything that might be measured. Statistical measures can guide the selection of predictors from a predetermined set of candidate predictors, but mathematics alone cannot direct the construction of that set. Ideally, as in all areas of practice, the selection of signals to be acquired and measures to be calculated should be driven by physiological hypotheses. For example, if it is hypothesized that the interaction of cognitive processes has been compromised following injury, then an ERP assessment based on a task such as the flanker arrow task [83] where the difficulty of two processes is manipulated (stimulus identification and response selection) might be disclosing. If impairment of the autonomic nervous system is suspected, measures of heart rate variability are indicated. Simply put, there is no substitute for clinical insight.

### 7. The Test–Retest Reliability of Psychophysiological Measures Is Frequently Untested and Can Be Unacceptably Low

*7.1. Observation*

While the deficiencies of psychophysiological measures in diagnosis are occasionally recognized, it is often argued that measures with low specificity can still be useful as longitudinal measures. A clear example is body temperature. A fever is not specific to a single disorder, but it is nonetheless an essential clinical measure, but body temperature is stable (reliable) in health and is responsive to disease progression or recovery. High test–retest reliability of a measure in a clinically stable population is essential to its use in longitudinal clinical assessment. The literature on the test–retest reliability of psychophysiological and neuropsychological measures is limited and discouraging. For example, Cole et al. [84] conducted a test–retest reliability study of four computerized neurocognitive assessment programs in a healthy active-duty military population. They concluded "However, overall test-retest reliabilities in four NCATs (Neurocognitive Assessment Tools) in a military population are consistent with reliabilities reported in the literature (non-military populations) and are lower than desired for clinical decision making." To consider the specific case of event-related potentials, on reflection, we should not be greatly surprised at their problematic test–retest reliability. Polich and Herbst [85] have identified over fifteen factors that can alter ERPs, including recent exercise, fatigue, and recent food consumption. Non-prescription drugs such as caffeine, nicotine, and alcohol, as well as prescribed medications, also alter ERPs. Outside of the research environment, in routine clinical practice it is very difficult to control for all of these factors. Polich and Herbst [85] do, however, present data indicating that the coefficient of variation of ERP measures are comparable to other medical tests, though it should be noted that the coefficient of variation is an imperfect measure of reliability.

*7.2. Response to Observation*

Estimating the test–retest reliability of a psychophysiological measure in healthy clinically stable controls is the essential first step because typically this population gives the best reliability (Gibson's Law). If a measure is not reliable in that population, its clinical utility is at best marginal. Unfortunately, methodological errors can be observed in some investigations of reliability. In the case of a continuous variable, which is typically the case for psychophysiological measures, linear product moment (Pearson) correlations do not provide an appropriate quantification of reliability. The intraclass correlation coefficient (ICC) is the appropriate measure. There are, however, several measures known collectively as intraclass correlation coefficients. Shrout and Fleiss [86] give six versions and McGraw and Wong [87] have ten versions. The choice depends on the reliability evaluation protocol being used. Müller and Büttner [88] and Koo and Li [89] provide selection guidance. Inappropriate versions of the intraclass correlation coefficient are often used. Because the numerical value of the ICC can vary considerably with the version being used, it is essential to include a specification of the ICC version in the report [90].

Interpretation of ICCs is problematic. Fleiss [91] described values in the range 0.4 to 0.75 as good to fair. Koo and Li [89] gave bands for four characterizations (poor, moderate, good, excellent). De Mast [92] has described generalizations of this kind as being "hopelessly arbitrary". Given this uncertainty, though often unreported, confidence intervals for ICCs are critical for their interpretation. A large literature has developed procedures for confidence interval construction for ICCs [86,93–95].

The interpretation of intraclass correlation coefficients can be further advanced by using them to calculate the standard error of measurement and the minimum detectable difference [96]. While helpful when interpreting ICCs, it should be understood that the standard error of measurement and the minimum detectable difference are statistical properties of distributions of the measure obtained in test–retest studies. They are not directly connected to the clinical response. The minimum detectable difference is not the same as the minimum *clinically*-important difference. Estimating the clinical importance of

a change in a psychophysiological variable requires additional validation. Anchor-based methods for connecting changes in psychophysiological variables to changes in clinical state are presented in Copay et al. [97]. Additionally, sample size estimates should be specific to reliability studies and are larger than often supposed. Zou [98] has derived sample size formulas for estimating intraclass correlation coefficients with precision and assurance. If the measure is being used longitudinally in a clinical trial, the test–retest interval of the validating study should be equal to the duration of the trial.

When possible, a test–retest reliability study of a new variable should incorporate the simultaneous measurement of another variable that is already known to be reliable under the conditions of the test in the population being tested, for example, simultaneous measurement of heart rate variability when evaluating a measure calculated from event-related potentials. If the new variable is found to be reliable, all is well. If the new variable is found to have low reliability, two possibilities exist: (1) the new variable is unreliable in these conditions with this population or (2) there were errors in the design and implementation of the reliability study. These two possibilities can be distinguished by observing the results obtained from the variable known to be reliable.

While establishing acceptable reliability in a healthy control population is the first step in the evaluation of a psychophysiological measure, it should be noted that reliability is population-specific. This is particularly true in neuropsychiatry. High variability is a characteristic of an injured or diseased central nervous system [99,100]. For example, a longitudinal change in a measure that would be remarkable in a healthy twenty-year-old might not be remarkable in a clinically-stable neuropsychiatric patient. Reliability should therefore be determined for the population of interest.

To summarize test–retest reliability requirements:

1. Test–retest reliability should be quantified with the intraclass correlation coefficient using an adequate sample size;
2. The version of the ICC used should be specified;
3. The report of the ICC should include confidence intervals and a specification of the procedure used to calculate the confidence interval;
4. The population used to determine the ICC should be appropriate for the clinical question being addressed;
5. Consideration should be given to including the simultaneous measurement of a variable of known reliability in order to evaluate the validity of the test–retest study;
6. The report should include determination of the standard error, the minimum detectable difference, and their confidence intervals;
7. If the measure is being used for pre- and post-trial evaluation in a clinical trial, the test–retest interval should be equal to the duration of the trial;
8. Consideration should be given to incorporating a determination of the minimum clinically-important difference into the study.

## 8. Variation of Psychophysiological Measures with Age, Sex, and Ethnicity

### 8.1. Observation

The task of establishing the reliability of psychophysiological measures is seen to be more demanding than might be supposed when it is noted that psychophysiological variables can show a dependence on age, sex, and ethnicity.

Measures of heart rate variability can show a dependence on age and gender (representative examples include: [101–104]), as can event-related potentials [105–107]. As will be described presently, gender differences in placebo-induced alterations of ERPs have been reported [108].

The literature describing ethnic and culture-specific variations in psychophysiological variables is smaller. Fukusaki et al. [109] found that gender-dependent HRV measures in Western populations were not observed in Japanese populations. While Choi et al. [110] found that age-dependent decreases in HRV measures commonly observed in Western populations were also observed in Korean populations, they did note that "The cause of

the difference in HRV depending on the gender between Westerners and Asians should be included in future studies." Mu et al. [111] found cultural differences in event-related potentials in Chinese and US populations in a social norm violation paradigm. Specifically, "the N400 at the frontal and temporal regions, however, was only observed among Chinese but not US participants, illustrating culture-specific neural substrates of the detection of norm violations."

### 8.2. Response to Observation

Operationally, these results indicate that when used longitudinally in a clinical study, the reliability of psychophysiological measures should be determined for the age, gender, and ethnicity of the study population. The procedures outlined in response to Observation 6 are applicable. If heterogeneous participant populations are used in a study, multiple determinations of the minimal detectable difference will be required.

## 9. Adaptation Not Repair: Psychophysiological Measures Do Not Invariably Normalize during Recovery

### 9.1. Observation

Consider the following scenario. A patient is diagnosed with a specific psychiatric disorder. Consistent with the prior literature, a psychophysiological measure obtained from this patient is found to be abnormal. Treatment is initiated and according to standardized patient-reported outcomes, clinical recovery is observed. In an ideal world, the psychophysiological measure would also normalize and, even better, track progress during the course of treatment. This is not an ideal world. As Steven Weinberg observed, "the universe was not designed to make physicists happy." Evidently this is also true for clinical psychophysiologists. Psychophysiological measures do not invariably normalize during recovery. This limits their utility in the longitudinal assessment of treatment. This is a potentially significant setback for clinical psychophysiology because the hope of utility in longitudinal assessment was deemed to be important when it was recognized that the nonspecificity of psychophysiological measures often precluded their use in diagnosis.

An inconsistent pattern is observed when the literature describing changes in psychophysiological measures in response to treatment is examined. Kemp et al. [112] found that HRV measures are lower in depressed patients and the magnitude of the decrease as compared to healthy controls was correlated with the clinically-perceived severity of the depression. This decrease was most apparent with nonlinear measures of HRV. Kemp, et al. report that tricyclic medication decreased HRV measures and that selective serotonin reuptake inhibitors (SSRIs), mirtazapine, and nefazodone had no significant effect on HRV even though patients responded clinically positively to treatment. Similarly, Brunoni et al. [113] found that depressed patients responded to sertraline or direct current electrical current therapy for depression, but lower HRV measures did not normalize. Bozkurt et al. [114] found no relationship between treatment response and change in HRV measures. The review by Alvares et al. [115] found reduced HRV in the patient groups considered (mood disorders, anxiety, psychosis, dependent disorders) and found that HRV did not normalize in response to successful medication. Psychotropic medication further reduced HRV specifically associated with tricyclic antidepressants and clozapine. In contrast, Udupa et al. [116] found that the form of treatment was an important consideration. In their patient population, HRV measures increased in response to rTMS, decreased in response to tricyclic antidepressants, and were effectively unchanged in response to SSRIs. The measures used to quantify heart rate variability may also be important. Nahshoni et al. [117] treated elderly patients presenting major depressive disorder with ECT. The pointwise dimension of HRV increased in responders and showed a tendency toward a correlation with symptom improvement. In this study, spectral measures of HRV, however, did not show a significant difference after ECT. This observation emphasizes the possible importance of developing comprehensive analysis protocols incorporating a large number of psychophysiological

measures. In the case of heart rate variability, the Kubios analysis suite is a significant contribution [118].

While, as previously noted there is a substantial literature describing alterations of event-related potentials in clinical populations, the literature describing treatment-dependent change or the absence of change in ERPs is much smaller. As in the case of measures of heart rate variability, the observations of longitudinal change in EEGs and event-related potentials during the course of treatment present a complex pattern of positive results (the measure normalizes in response to successful treatment) and negative results (the measure does not normalize). In a study of depression and of schizophrenia, Buchheim et al. [119] in a clinical study of depressed patients found that psychodynamic psychotherapy results in a normalization of the late positive potential (LPP) and gamma abnormalities that had previously been identified by Siegle et al. [120]. Decreased amplitude of the P3 observed in medication-free depressed patients normalized after four weeks of antidepressant treatment [121]. Similarly, P3 amplitude increased in response to electroconvulsive therapy [122].

A meta-analysis by Umbricht and Krljes [123] concluded that defects in the mismatch negativity (MMN) ERP are "a robust feature in chronic schizophrenia", and it had been hypothesized that glutathione dysregulation and subsequent N-methyl-D-aspartate hypofunction may be an element in the pathophysiology of schizophrenia. Responding to this hypothesis, Lavoie et al. [124] administered N-methyl-cysteine, a glutathione precursor, to schizophrenia patients and observed that treatment significantly improved MMN generation, without measurable effects on the P300. MMN improvement was "observed in the absence of robust changes in assessments of clinical severity, though the later was observed in a larger and more prolonged clinical study." Similarly, Zhou et al. [125] found that treatment of schizophrenic patients with aripiprazole improved amplitude of the MMN and reduced Positive and Negative Syndrome Scale scores.

The results with ERPs just cited contrast with results obtained in the treatment of anxiety. Error-related negativity (ERN) is an ERP component observed at frontal and central electrodes after the participant makes an incorrect response (reviewed in Gehring et al. [126]). Anxiety disorders are associated with an increased amplitude of the ERN [127,128]. Successful pharmacological treatment of anxiety does not, however, normalize the ERN [129–133]. Valt et al. [134] investigated a related internalizing disorder, panic disorder, and, consistent with prior literature, found that compared to controls, untreated participants had a greater ERN and also a greater vertex positive potential. In a subsequent treatment study, Valt et al. [135] found that, as before, the ERN did not normalize in response to psychological treatment, but treatment-related normalization of the vertex positive potential was observed.

*9.2. Response to Observation*

In the specific context of anxiety disorders, Hajcak et al. [136] raise the following interesting possibility: "The evidence suggests that typical treatments for anxiety do not normalize an increased ERN. One possibility is that the ERN is related to the risk for anxiety but not the expression of an anxious phenotype. In this case, treatment-related effects on the ERN would not be expected unless treatments alter underlying risk processes that are reflected in the ERN." Considered more generally, in those presentations where psychophysiological measures are indicative of risk and not the presence of disease, the utility of these measures as longitudinal markers of treatment progress will be limited.

In the case of traumatic brain injury, observations of recovery of function in the absence of normalization of psychophysiological variables are frequently observed. The work of Kurt Goldstein [137] is of particular interest. Goldstein treated patients who had sustained significant brain injuries during World War I and concluded that in many cases when restoration of function is observed, the injured brain is not repaired; it adapts. Goldstein focused on traumatic brain injury. It seems possible that the principle of adaptation, not

repair, is more generally applicable in neuropsychiatry. Restoration or partial restoration of function in the absence of normalization of clinical markers may be common.

In those instances where either of the two possibilities just considered—(1) psychophysiological measures are indicators of risk, not disease state, or (2) adaptation in the absence of repair—are correct, a simple response to Observation 8 that psychophysiological measures do not invariably normalize in response to a clinically successful treatment will not be available.

An alternative treatment strategy utilizing psychophysiological measures of considerable clinical value may be available. In typical clinical practice, treatments are directed to the resolution of symptoms. It is possible to consider treatment protocols explicitly directed to the normalization of aberrant psychophysiological measures. This has been addressed by Hajcak et al. [136]. In their summary, the focus is on error-related negativity, but the concept is more generally applicable: "Traditional CBT (cognitive behavior therapy) and SSRIs do not seem to impact the ERN, and the ERN does not appear to robustly predict treatment response to traditional CBT or SSRIs. Yet the ERN is a robust predictor of subsequent changes in symptoms and psychopathology, and a range of strategies appear to modulate the ERN, at least in the short term. To us, these data collectively suggest the need to test and develop novel interventions that are more focused on altering the ERN. To our knowledge, no intervention has been designed to directly target the ERN, and pharmacological studies have not routinely considered ERPs as potential targets. Moreover, brain stimulation and neurofeedback might provide additional and more direct methods of altering ERPs."

The treat-the-biomarker strategy is the organizing principle of many neurofeedback protocols [138] such as theta/beta ratio training in attention deficit hyperactivity disorder (ADHD) [139], and frontal alpha asymmetry in depression [140,141]. See, however, Papo [142] for a review of unresolved issues in neurofeedback. Similarly, HRV biofeedback seeks change in a physiological variable, not a change in a behavior. The role of psychophysiology in clinical practice may increase substantially as more disorder-specific psychophysiological signatures are identified and incorporated into neurofeedback protocols.

## 10. Psychophysiological Biomarkers Can Change and, in Some Cases, Normalize in Response to Placebo Interventions

### 10.1. Observation

Placebos can alter measures of heart rate variability, eye tracking behavior, and event-related potentials. Vaschillo et al. [143] measured the impact of alcohol and placebo on measures of heart rate variability obtained during an emotional cue challenge. Participants were presented with pictures (negative, positive, neutral) in a 5 s on/5 s off protocol (0.1 Hz). Three commonly used measures of heart rate variability were recorded (standard deviation of interbeat intervals, the percentage of interbeat intervals exceeding 50 ms, and high-frequency heart rate variability). Additionally, they reported a novel measure, the 0.1-Hz index, which is the maximum amplitude of the interbeat interval spectrum in the 0.075 to 0.108 Hz range. The 0.01-Hz index was diminished by both alcohol and placebo, and there was no statistically significant difference between the alcohol and placebo group as determined by this measure. Vaschillo et al. concluded that this suggests a dependence of this measure of heart rate variability on the participant's cognitive expectancy. Darragh et al. [144] studied placebo modification of heart rate variability during the recovery from cognitive stress. An acute reduction in heart rate variability was induced by an arithmetic test conducted in the presence of a research assistant. The experimental group was administered a placebo nasal spray and was told that it contained serotonin and that it was expected that this would accelerate recovery from stress. Two metrics of heart rate variability were employed: high-frequency spectral power and the root mean square of successive differences. An increase in vagally-mediated heart rate variability was observed in the placebo treated group contra the untreated control group.

Daniali and Flaten [145] conducted a systematic review of the effects of placebo analgesia and nocebo hyperalgesia on cardiac activity. Specifically, they reviewed papers

that reported blood pressure, heart rate, and heart rate variability. They identified six papers that reported effects on heart rate variability and provide the following summary: "The results indicate that the placebo analgesic effect is associated with a decrease in LF-HRV (low frequency heart rate variability), that the nocebo hyperalgesic effect is associated with an increase in LF-HRV, and that HRV is a predictor for placebo effects. However, there is no reliable effect of placebo on the LF/HF (low frequency/high frequency) ratio and HF-HRV."

Schienle et al. [146] found that placebos can affect eye tracking behavior. In this experiment participants viewed neutral and disgust-inducing images with and without a "disgust placebo", an inert pill presented with the verbal suggestion of disgust relief. In an experiment in which participants looked at side-by-side images of neutral–disgust and neutral–neutral images, it was found that the placebo resulted in a marked decrease in reported distress and an increased number of fixations on disgusting images. In similarly designed fMRI experiments, Schienle et al. [147,148] found that the disgust placebo reduced the activation of the insula and the visual cortex and reduced experienced disgust. In an eye tracking experiment investigating placebo effects on phobias, specifically spider phobia [149], participants were shown spider pictures paired with neutral pictures with and without a placebo labeled as propranolol. Fixation count and dwell time increased in the placebo condition, and there was a slight decrease in reported symptom severity.

Placebo interventions can also alter event-related potentials. Placebo effects on ERPs have been observed in studies of analgesia, anxiolysis, emotional processing, and cognitive enhancement. Building on a substantial prior literature investigating the effects of placebo analgesia on ERPs [150,151], Aslaksen et al. [108] investigated the effects of a placebo intervention in a pain study where ERPs were evoked by heat pulses. The amplitude of the N2/P2 complex (the amplitude difference between the negative going N2 ERP component and the positive P2 component elicited in this experiment by a heat stimulus) was reduced by the placebo. Interestingly, a reduction in pain reported and the P2 amplitude was observed at a group level in the male participant group, but not in the female participant group.

In a study reported by Meyer et al. [152], inactive treatment was accompanied by verbal suggestion that the placebo intervention would have an anxiolytic effect of experimentally-induced phobic fear or sustained anxiety. A placebo-dependent sustained increase of frontal midline EEG theta power and an increase in frontoposterior theta coupling consistent with activation of cognitive control mechanisms was observed. Downregulation of unspecific cue reactivity was observed in fear ratings, skin conductance response, P300 amplitude (280–400 ms), and the late positive potential (400–700 ms).

A complex mix of results have been obtained in ERP studies of processing of emotionally-valanced images with and without a placebo intervention. Übel et al. [153] studied the processing of emotionally-valanced images by children. Children viewed disgusting fear-eliciting images and neutral images with and without placebo (syrup presented with the verbal suggestion that it would ease disgust symptoms). In this experiment, the placebo increased the late positive potential (defined here as 400–1000 ms) in response to disgusting and fear-eliciting photographs. Übel et al. propose the following interpretation: "These findings suggest that the placebo had the function of a safety signal which helped the children to direct their automatic attention to the aversive stimuli and to overcome visual avoidance." In Schienle et al. [154], an unpleasant context, in this study a bitter after-taste elicited by wormwood tea administered prior to ERP recording, could reduce the late positive potential elicited by affectively-valanced pictures. In this study, there were three experimental groups: water instead of tea, tea, and tea and a placebo treatment: light therapy on the tongue to "reduce sensitivity of taste buds". Two classes of pictures were shown: neutral and disgusting. For both classes of pictures, the early late positive potential was smaller in the tea/no placebo case than in the case water case, with the tea/placebo amplitudes being intermediate to them. The authors state, "This is the first EEG study

to demonstrate effects of a context-targeting placebo", the context being the wormwood tea pretreatment.

The possibility of using transcranial direct current stimulation (tCDS) to enhance cognitive performance has received excited public attention. Van Elk et al. [155] conducted an ERP study where the placebo was an inert sham tCDS. Participants were advised that their performance would be enhanced in tCDS trials. Participants reported improved subjective performance during placebo enhancement, but objective performance was unchanged. During the induction phase, placebo-induced expectation of enhancement increased frontal theta power "potentially reflecting a process of increased cognitive central allocation". The placebo manipulation did not, however, change the ERN associated with incorrect responses.

One of the most intriguing studies of placebo impact on ERPs is the Guevarra et al. [156] study utilizing a "non-deceptive" placebo. Non-deceptive placebos are not original to this study [157,158]; additionally, Guevarra et al. cite studies describing the beneficial effects of non-deceptive placebos for several disorders, but insofar as we know, the Guevarra et al. study is the first study showing that non-depictive placebos can change ERPs. They give the following description of the non-deceptive placebo protocol used in their study: "Participants in the non-deceptive placebo group read about placebo effects and were then asked to inhale a nasal spray consisting of saline solution. They were told that the nasal spray was a placebo that contained no active ingredients, but would help reduce their negative emotional reactions to viewing distressing images if they believed it would. Participants in the control group read about the neural processes underlying the experience of pain and were also asked to inhale the same saline solution spray; however, they were told that the purpose of the nasal spray was to improve the clarity of the physiological readings we were recording in the study. The articles were matched for narrative structure, emotional content and length."

The Guevarra et al. experiment was a negative picture viewing task. In the first experiment, participants viewed a picture (neutral or negative image) and were asked "Rate how does this picture makes you feel"? on a one-to-nine scale. The non-deceptive placebo reduced self-reported measures of emotional distress. In the second experiment, the participants viewed neutral and negative images as before, but were not asked to rate them. ERPs were recorded in the second experiment. The ERP analysis of their study focused on the late positive potential (LPP). Two distinct components have been identified in the LPP and they have different cognitive associations. The early component, with a latency of 400–1000 ms, corresponds to attention allocation [159]. The late component, with a latency of 1000–1600 ms, is associated with conscious appraisal and emotional processing [159,160]. The late positive potential is down-regulated by cognitive emotion-regulation strategies [161]. As reported above, some studies suggest that deceptive placebos amplify attention to negative stimuli and other studies suggest the opposite, where attention is quantified by the amplitude of the early component of the LPP. Guevarra et al. found no reliable non-deceptive placebo effect on the early LPP. In contrast, however, they did observe that the non-deceptive placebo reduced activity during the later component of the LPP, which quantifies "meaning-making stages of emotional reactivity".

### 10.2. Response to Observation

The studies showing an effect of placebos on psychophysiological measures summarized here are laboratory investigations conducted on a time scale of hours. The degree to which they are applicable to clinical trials conducted over a period of weeks or months is unclear. It is commonly suggested that placebo effects are short-lived, while clinically-induced physiological change is longer lasting. If this is true, and this view is subject to challenge, it might follow that psychophysiological measures will be useful in long-term follow-up. Investigations of the long-term impact of placebos on psychophysiological measures are warranted. In any case, we must be alert to the possibility that, as with patient-reported outcomes (PROs), physiological measures will be an imperfect measure

of treatment response. The results summarized by Daniali and Flaten [145] indicate that more than one measure of heart rate variability should be examined. While they report placebo and nocebo changes in low frequency HRV, there was no reliable effect on the low-frequency to high-frequency ratio. Physiological measures should be used in combination with PROs. Instances where the effect size calculated with PROs is small while the effect size calculated with a measure of HRV is large might be taken to indicate a physiological treatment response that was not captured in a PRO report.

The magnitude of a placebo response can be estimated using a placebo control group. As an additional check, it is possible to ask if active treatment responders and placebo responders are different at intake. If pre-treatment measures that characterize active treatment responders are indistinguishable from measures that characterize placebo responders and if the frequency of active versus placebo response is nearly equal, then perhaps the active treatment response is a placebo response.

Determination of the magnitude of either a placebo response or a treatment response can be confounded by spontaneous recovery. It is commonly recognized that spontaneous recovery can be a significant complication in depression treatment trials. Indeed, Kraepelin (cited by Posternak et al. [162]) concluded that untreated depressive episodes would last six to eight months. The frequency of spontaneous recovery can be estimated from a waitlist control group. Posternak and Miller [163] collected and analyzed results obtained in antidepressant trials that included a waitlist control group. They provided the following summary: "Our analysis of 19 studies involving 221 depressed subjects randomized to a waiting list for 2–20 week found a mean decrease in symptomatology of 10–15%. A sub-analysis of 11 studies that obtained depression rating scores between weeks 4 and 8—the time frame used in most antidepressant trials—yielded similar results. We therefore would postulate that subjects enrolled in short-term antidepressant trials probably improve on their own by this amount." In a subsequent study Posternak et al. [162] found that depressed patients who went without somatic therapy throughout the course of a depressive episode presented a median episode duration of 13 weeks.

The possibility of both a placebo intervention altering psychophysiological measures and the possibility of spontaneous recovery argue for the incorporation of both waitlist and placebo arms in studies investigating the longitudinal response of psychophysiological measures to treatment.

## 11. The Mathematical Procedures of Statistical Learning Are Not Robust to Misapplication and to Data Artifacts

*11.1. Observation*

Two related final concerns merit attention: the impact of data artifacts and the misapplication of analysis algorithms. Increasingly sophisticated forms of analysis are increasingly sensitive to artifacts in data. Muthukumaraswamy [164] noted that "As analytical techniques in MEG/EEG analysis . . . become increasingly complex, it is important that artifact-free data are being fed into these algorithms." Experience suggests that this is particularly true of methods that quantify nonlinear structure in experimental data. It has been shown, for example, that filtered noise can mimic low-dimensional chaotic attractors [165]. These observations have led to a re-examination of evidence of low-dimensional structure in the EEG [166]. For the specific case of gamma band EEG activity, Hipp and Siegel [167] and Muthukumaraswamy [164] provide practical guidelines for investigations of gamma band activity. Luck [168] offers a comprehensive guide to artifact detection in ERP studies. He also suggests cautions concerning artifact correction. We concur and in particular note that artifact correction procedures can distort results obtained with sensitive measures of information movement.

In addition to concerns about data quality, it should be understood that analytical methods are not robust to misapplication. Three failure patterns are particularly prominent: (1) p-hacking, (2) errors in the application of signal processing algorithms, and (3) errors in the application of statistical learning algorithms. Errors in statistical analysis, volitional

or inadvertent, collectively known as p-hacking (which includes multiple comparisons in hypothesis testing, over-hacking (the practice of continuing to manipulate the data to obtain a p value lower than 0.05), selection bias in the choice of analysis pathways, and selective debugging of analysis software) are well documented [169,170], but nonetheless occur in the peer reviewed literature.

Analysis errors in signal analysis includes errors in the procedures used to construct random phase surrogates that can result in the false-positive indication of deterministic structure in random data [171]. Similarly, inappropriate procedures for calculating the complexity of a time series can also give false positive indications of deterministic structure [172]. As noted in the introduction to this contribution, the three primary objectives of clinical psychophysiology are diagnosis, the longitudinal assessment of treatment response, and the identification of individuals at risk of disease onset. All of these objectives are classification problems. Cross-validation is an essential step in confirming classification results. It is, however, easily misused. An incorrectly constructed cross-validation calculation can seemingly validate a classification procedure constructed with random numbers. Indeed, the error is so commonly encountered that the standard textbook on the subject of statistical learning [81] includes a section with the title "The Wrong and Right Way to Do Cross-Validation". With the increasing availability of powerful and freely accessible statistical learning software tools, errors of this kind will almost certainly become more common.

*11.2. Response to Observation 10*

In 2019, the Society for Psychophysiological Research conducted a one-day workshop on open science to addresses some of the issues raised here. Their results were published in Garrett-Rufin et al. [173]. The panel's recommendations included data sharing with data format harmonization, analysis pipeline sharing, pre-registration of an analysis plan, multisite studies, and encouraging replication studies. The resources of the Open Science Framework (Center for Open Science, [174]) can facilitate this effort. Saunders and Inzlicht [175] have outlined procedures that can increase the transparency of meta-analyses.

The Society for Psychophysiology call for standardization was not the first to stress the need for standardization in order to increase the clinical utility of psychophysiological biomarkers [60,176–180]. The most comprehensive effort to date has been the development of ERP CORE by Luck and his colleagues [181]. They have made freely available "a set of optimized paradigms, experiment control scripts, data processing pipeline and sample data (N = 40 neurotypical young adults) for seven widely used ERP components". These authors and others have noted that rather than use a generic two stimulus P3 oddball protocol, the ERP paradigm used should be consistent with the clinical presentation under consideration. Representative examples could include reduced contingent negative variation in ADHD [182], reduced mismatch negativity in schizophrenia [123], absence of P50 (the second positive component of the ERP elicited in a double click protocol) suppression in schizophrenia [183], decreased P3b (the ERP activity associated with attention and memory processing) in depression [184], enhanced error related negativity in anxiety disorders [132], pattern separation/pattern completion in dementia [185], and increased error related negativity in obsessive–compulsive disorder [129]. Further, increased clinical utility of ERPs may be obtained with the expansion of ERP analysis beyond an examination of amplitude and latency of averaged ERP waveforms with the incorporation of measures of synchronization [186], event related oscillations [187], microstates [188], information dynamics [189], and network analysis [190].

As is generally recognized, statistical tests should be incorporated routinely in psychophysiological studies, though as noted above, important tests such as surrogate data calculations and cross-validation can be misused. Running all analysis software against publicly-available standard data sets provides an important mechanism for validating the software used in a specific study. Bootstrapped confidence intervals [191,192] for reported psychophysiological measures can be an important safeguard. Wang et al. [69] provide an illustrative example. In a search for electrophysiological prodromes of delayed

onset PTSD, initial calculations indicated sensitivity and specificity of approximately 0.8. The corresponding confidence intervals, however, were found to be [0, 1], thus indicating that the initial encouraging result was the fortuitous artifact of a small sample size. The previously-cited analysis of Button et al. [73] concerning the dangers of underpowered studies is relevant.

In the case of the classification calculations, which are essential to implementing the three previously stated objectives of clinical psychophysiology, Fernández-Delgado et al. [193] offer guidance on the choice of a classification algorithm. The OpenML website [194] provides access to experimental data including data from the University of California, Irvine machine learning database [195] that can be used in pipeline validation studies. For confounds specific to EEG classification, see Li et al. [196] and Ahmed et al. [197].

## 12. Discussion

Clinical psychophysiology has three primary objectives: (1) diagnosis, (2) longitudinal assessment of treatment response or disease progression, and (3) identification of physiological prodromes that can identify individuals at risk of clinical presentation with sufficient confidence to warrant preemptive treatment. It has been proposed that biomarkers constructed from psychophysiological measures can contribute to this effort. The objective of this study was to identify challenges to this program and, when possible, to identify strategies to address these challenges.

The concept of a biomarker itself must be examined. Within the psychophysiological community, and indeed in this paper, the term biomarker is used casually. The US Food and Drug Administration has issued technical definitions of biomarkers [198] and has issued draft criteria for biomarker qualification [199,200]. That guidance states: "Qualifiation of a biomarker is a determination that within the stated context of use, the biomarker can be relied on to have a specific interpretation and application in drug development and regulatory review." Many proposed biomarkers fail to meet the standards set by these more rigorous definitions. The report by Prata et al. [201] specifically focusing on schizophrenia is instructive. They provide the following summary: "We categorized all PubMed-indexed articles investigating psychosis-related biomarkers to date (over 3200). Fewer than 200 studies investigated biomarkers longitudinally for prediction of illness course and treatment response. These biomarkers were then evaluated in terms of their statistical reliability and clinical effect size. Only one passed our a priori threshold for clinical applicability." The single biomarker that met the Prata et al. criteria is a single nucleotide polymorphism predicting risk for clozapine-induced agranulocytosis.

Subsequent research suggests, however, that a more optimistic assessment may be appropriate. Of many promising studies, we make note of two based on serum biomarkers. Chan et al. [202] presented results obtained from a serum biomarker constructed to discriminate between depressed patients and healthy controls. A 10-fold cross validation and application of a least absolute shrinkage and selection operator (LASSO) resulted in an optimal panel of 33 immune-neuroendocrine biomarkers and gender. Moderate-to-good discrimination was observed in terms of area under the curve (AUC), with a 95% confidence interval for the AUC of $(0.62, 0.86)$. Recall that a classification no better than chance gives an AUC of 0.5. In the case of first episode patients that were free of chronic non-psychiatric illness with the incorporation of demographic covariates, the AUC improved, giving a 95% confidence interval of $(0.76, 0.92)$.

Chan et al. [203] compared 127 first onset drug naïve schizophrenia patients with 204 controls. Using LASSO, they identified 26 biomarkers that best discriminated between patients and controls. This was validated against two independent cohorts. The schizophrenia detection study gave $0.95 < \text{AUC} < 1.00$ (95% confidence intervals for the AUC). The predictive performance was tested in a pre-onset population where $0.86 < \text{AUC} < 0.95$. In a prodromal population, the biomarker panel gave $0.71 < \text{AUC} < 0.93$, which was improved to $0.82 < \text{AUC} < 0.98$ by incorporating the positive symptom subscale of the Comprehensive Assessment of At-Risk Mental State.

It should be noted, however, that these encouraging results [202,203] were obtained from carefully constructed clinical studies comparing either depressed patients versus controls or schizophrenics versus controls. The performance in general psychiatric practice, which would include a mix of these patient populations along with generalized anxiety disorder, PTSD, bipolar disorder, and others, is unclear.

The role of psychophysiology in advancing our understanding of central nervous system physiology is clear. We have, however, been addressing a different question: what are the prospect of the near-term utility of psychophysiological measures in clinical practice? The analysis presented here suggests several steps that could accelerate the introduction of psychophysiological measures into clinical practice. Several were recommended in the 2019 Society for Psychophysiology meeting [173] and have been implemented in the ERP CORE project [181].

1. Common inclusion/exclusion criteria should be used to define study populations. A lack of uniformity in defining, for example, depression has made it impossible to compare results obtained in different studies. These criteria should be based on standardized questionnaires (patient-reported outcomes) that satisfy the COSMIN Criteria;
2. Standardized data acquisition protocols should be used;
3. Explicit descriptions of data analysis procedures and ideally analysis software should be provided;
4. Uniform result report formats should be adopted;
5. Deidentified data should be publicly available for independent reanalysis. In addition to raw physiological data from each participant, this availability should include the item-by-item questionnaire results that established study eligibility. This will make it possible to correlate specific elements of the clinical presentation with measures calculated from the physiological data.

Overall, our assessment for near term utility of psychophysiology in clinical practice is one of guarded optimism. The effort to introduce these measures into practice will be a journey well worth traveling, but may be a longer journey than might be supposed.

**Author Contributions:** P.E.R.: conceptualization, literature review, first draft; C.C.: revision of the final draft; D.D.: review of statistical commentary; D.K.: conceptualization and revision of the final draft. All authors have read and agreed to the published version of the manuscript.

**Funding:** This research received no external funding.

**Institutional Review Board Statement:** Not applicable: This contribution is a review of publicly accessible previously published literature.

**Informed Consent Statement:** Not applicable.

**Data Availability Statement:** Not applicable.

**Acknowledgments:** PER would like to acknowledge Adele Gilpin and the students, faculty, and staff of the Center for the Neurobiology of Learning and Memory at the University of California, Irvine and its director, Michael Yassa, for discussion critical to the development of this contribution.

**Conflicts of Interest:** The authors declare no conflict of interest.

**Disclaimer:** The opinions and assertions contained herein are those of the authors and do not necessarily reflect the official policy or position of the Uniformed Services University, the Department of Defense, or the Henry M Jackson Foundation for the Advancement of Military Medicine.

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
