# Peer review of "Cautionary Observations Concerning the Introduction of Psychophysiological Biomarkers into Neuropsychiatric Practice"

_2673-5318, doi:10.3390/psychiatryint3020015_

Round 1

Reviewer 1 Report

To the authors

Main comments

The question of mechanisms regulating consciousness and modern artificial intelligence techniques such as machine learning approaches is welcome in physiology. Probably the use of these techniques can be additionally useful in psychobiological variables where variability is a peculiar issue in these outcomes. 

The study is interesting, but it is almost impossible to follow the reading of the article because the text has a writing model and general structuring very far from a scientific article. My main criticism of the article is the lack of reporting and structuring as follows:

- in the abstract inform the type of study, and clearly (and explicitly) inform the study objective and the conclusion of the study in a coherent manner;

- in the introduction bring what is already known and make the reader clearly understand what is known, and soon after, inform what is not yet known.

- state the chapters of the paper in IMRaD sequence.

- Indicate at the end of the introduction what the study objectives

- answer the objectives of the study briefly at the beginning of the discussion

- conclude with the main findings and interpretation of the study.

- the figure 1 is missing.

- Many sentences need references. Consider inserting some references as suggested below

I suggest inserting examples of machine learning techniques used generally in psychophysiology (e.g. PMID: PMID: 35314708, 33331795, PMID: 34693282 , PMID: 35317343).

I suggest inserting at the beginning of the introduction the meaning of energy efficiency in a larger context to support the analysis of the role of psychophysiology variables in the specific function more robustly, so I suggest inserting the following article (PMID: 30618802) that carries the concept of efficiency. And specifically, I suggest the following wording of the sentence in lines 16-19:

" Three applications are critical to this program: diagnosis, longitudinal assessment of treatment response or disease progression, and identification of individuals in the subsyndromal state who are at risk of neuropsychiatric disorders. And, thus, these applications may impact in the overall metabolic efficiency [PMID: 30618802]. "

The strengths of the article are:

- study question

- writing.

Reviewer 2 Report

Thank you very much for the opportunity to review this very interesting article.

Overall, it is a well-written clear and concise article providing a good insight into the advantages and limitations of the application of biomarkers into neuropsychiatric practice.

I believe that the readership will find it most useful.

Reviewer 3 Report

I appreciate the opportunity to review this manuscript which proposes 10 cautionary observations and suggestions for remediation concerning psychophysiological information into neuropsychiatric practice. Overall, this is a very thought-provoking manuscript that seems well-timed with the evolution of machine learning and big data sets.

There are a number of suggestions and comments the authors may wish to consider to further enhance their manuscript.

The introduction seems a bit underdeveloped after reading the paper as a whole. It would be helpful if the authors could provide clearer statements of purpose and aims so that the reader has a clear idea of what they are about to learn, and why it is important. Some of these ideas are here, though the introduction seems overly concise. This might also provide more of a framework to develop a discussion section that more broadly reflects the paper as a whole and provides more integrated analysis/summary of the points the authors are making and illustrating (see comment about discussion section below).

Under response to observation 1, did the authors consider symptom validity assessment as part of patient report? In addition to using standardized and validated questionnaires for patient report, considering the validity of that report could perhaps add value.

Under observation 2, I would appreciate an additional clarification of how the term “consciousness” is being used. The authors provide a brief mention of higher cognitive processes, though from my standpoint the term consciousness to reflect cognitive function is unconventional and I would appreciate additional explanation. Under response to observation 2, it might be worth adding examples with citations to illustrate the “starting points” the authors refer to.

Observation 3 seems underdeveloped. It would be helpful to have explanation on what the authors mean by unconscious processes. This sounds a bit psychoanalytic in nature, though I believe the authors are intending to reflect measurable processes so further elaboration would be of interest. Similarly, under response to observation 3 further elaboration/examples of unconscious processes that are quantitatively measured would provide greater insight to the reader in terms of understanding the point the authors are making.

Observation 8 provides a great deal of detail about HRV variability under different treatment conditions. While insightful, the extensive examples to illustrate their point seems out of place among numerous other areas where points are made without much (or any) illustration.

The discussion section does not seem to entirely reflect an integration of observations and responses made in previous sections. The point is made that biomarkers are poorly defined, and then a few examples are provided of more rigorously tested biomarkers. This feels overly specific and perhaps incomplete in terms of an integrated analysis and discussion of the overall points the authors are trying to make.

The authors should spell out all acronyms at first mention.

The authors might consider some sort of graphic/table to help break up text and provide summary of major points. This is a relatively dense and long article and would benefit from opportunities to summarize key points.

Round 2

Reviewer 1 Report

Main comments

The authors aimed to identify challenges to use stats technologies combined with large databases of psychophysiological in the psychiatric clinical setups. The authors observed:

# The no specificity of psychophysiological measures complicates their use in diagnosis.

# Low test-retest reliability complicates use in longitudinal assessment, and quantitative psychophysiological measures can normalize in response to placebo intervention.

# Ten cautionary observations were presented.

I found some minor points, the discussion is important to the clinical context.

Minor points

Lines 1-2:

The combination of statistical learning technologies with large databases of psychophysiological data  has appropriately generated enthusiastic interest in future clinical applicability

Please, consider including the specificity of psychiatric area here.

I miss the citation on usage of physical-functional markers related to neuropsychiatric disorder as quantity of physical activity, 10m walking test. There are a myriad of studies showing high predictive value with low physical activity and functional mobility to some psychiatric disorder.

A conceptual model summarizing the challenges for this approach is missing.

Author Response

See attached file Response to Reviewer 1 Review of Version 2
